# Relationship between Air Temperature Parameters and the Number of Deaths Stratified by Cause in Gifu Prefecture, Japan

**DOI:** 10.3390/healthcare8010035

**Published:** 2020-02-07

**Authors:** Masaki Bando, Nobuyuki Miyatake, Hiroaki Kataoka, Hiroshi Kinoshita, Naoko Tanaka, Hiromi Suzuki, Akihiko Katayama

**Affiliations:** 1Department of Hygiene, Faculty of Medicine, Kagawa University, Miki-cho, Kagawa 761-0793, Japan; miyarin@med.kagawa-u.ac.jp (N.M.); tanzuki@med.kagawa-u.ac.jp (H.S.); 2Department of Physical Therapy A College of Kenshokai Academy, Tokushima city, Tokushima 760-0093, Japan; 3Department of Physical Therapy, Okayama Institute for Medical and Technical Sciences, Okayama city, Okayama 700-0913, Japan; h.kataoka59@gmail.com; 4Department of Forensic Medicine, Faculty of Medicine, Kagawa University, Miki-cho, Kagawa 761-0793, Japan; kinochin@med.kagawa-u.ac.jp (H.K.); ntanaka@med.kagawa-u.ac.jp (N.T.); 5Faculty of Social Studies, Shikokugakuin University, Zentsuji city, Kagawa 765-0013, Japan; kata@med.kagawa-u.ac.jp

**Keywords:** air temperature, the number of deaths, accidents, renal failure

## Abstract

Objective: It is well known that air temperature is closely related to health outcomes. We investigated the relationship between air temperature parameters and the number of deaths stratified by cause in Gifu prefecture, Japan. Methods: The number of deaths stratified by cause in Gifu prefecture Japan between January 2007 and December 2016 was obtained from the official homepage of Gifu prefecture, Japan. Air temperature parameters (°C), i.e., the mean air temperature, mean of the highest air temperature, mean of the lowest air temperature, the highest air temperature, and the lowest air temperature during the same period in Gifu city were also obtained from the Japan Meteorological Agency official home page. The relationship between air temperature parameters and the number of deaths was evaluated in an ecological study. Results: The number of deaths due to heart disease, cerebrovascular disease, pneumonia, accidents, or renal failure in January (coldest winter season in Japan) was the highest among the months. Simple correlation analysis also demonstrated a significant and negative relationship between air temperature parameters and the number of deaths due to heart disease, cerebrovascular disease, senility, pneumonia, accidents, and renal failure. Conclusion: Lower air temperature may be associated with a higher number of deaths due to diseases in Gifu prefecture, Japan.

## 1. Introduction

Many studies in the literature showed that environmental factors, including humidity [1], daylight hours [2], and air pollutions [3], are associated with health outcomes. In addition, it is well known that meteorological parameters, especially air temperature, are closely related to health outcomes, i.e., heart disease [4,5,6,7,8,9], cerebrovascular disease [10,11], pneumonia [12,13,14,15], and others [16]. In Japan, there are four seasons, including spring, summer, autumn, and winter, resulting in meteorological flections. We previously reported a significant relationship between air temperature and the total number of deaths, but not the number of deaths stratified by causes in Sakata city [17], Takamatsu city [18], Asahikawa city [19], and all 47 prefectures of Japan [20]. The number of ambulance transports was found to be associated with air temperature, and lower air temperature (winter season) was also closely associated with a higher total number of deaths [17,18,19,20]. Therefore, an effective strategy to prevent deaths affected by air temperature is required in Japan.

Air temperature has been reported to have a clinical impact on heart disease [4,5,6,7,8,9], cerebrovascular disease [10,11], and pneumonia [12,13,14,15]. However, many of these studies, including our previous reports, investigated the relationship between specific diseases and metrological parameters. In addition, many of these studies were performed in foreign countries [4,5,6,7,8,10,11,12,13,14,15]. Thus, few studies have evaluated the relationship between air temperature parameters and the number of deaths stratified by cause in Japan [21]. If we clarify these relationships, possible strategies to prevent deaths affected by lower air temperature could be recommended in Japan.

In this ecological study, we evaluated the relationship between air temperature parameters and the number of deaths stratified by cause in Gifu prefecture, Japan.

## 2. Methods

### 2.1. Study Area

Gifu prefecture, which is centrally located in Japan and surrounded by mountains, has over 2,000,000 people living in an area of approximately 10,621 km^2^. The population density was approximately 191 people/km^2^ in 2015 [22].

### 2.2. Number of Deaths

The number of deaths stratified by cause in Gifu prefecture, Japan, between January 2007 and December 2016, was obtained from the official home page of Gifu prefecture, Japan [23]. We used the total number of deaths in all ages due to malignant neoplasm, heart disease, cerebrovascular disease, senility, pneumonia, accidents, including traffic accidents and others, aspiration pneumonitis, renal failure, and vascular and unspecified dementia, according to the rank of the entire Japanese population in 2018 [24]. However, we were unable to include the number of deaths due to aspiration pneumonia because the collection method changed in 2017. The population during the same period in Gifu prefecture [23] was also used to adjust the number of deaths. The number of deaths stratified by cause, which was adjusted by population and the number of days per month, was used for analysis.

### 2.3. Air Temperature Parameters

Air temperature parameters (℃), i.e., the mean air temperature, mean of the highest air temperature, mean of the lowest air temperature, the highest air temperature, and the lowest air temperature during the same period in Gifu city, were obtained from the Japan Meteorological Agency official home page [25]. The observation spot was located almost in the center of Gifu prefecture.

### 2.4. Ethics

For this ecological study, the number of deaths stratified by cause and the air temperature parameters were obtained from the official homepage. The ethics committee of Shikoku Gakuin University approved this study (approval number: 2019003, approval date: 11 December 2019).

### 2.5. Statistical Analysis

Data were expressed as the mean ± standard deviation (SD). The number of deaths by month was compared using the Kruskal–Wallis test and Steel test, with *p* < 0.05 being considered significant. The relationships between the number of deaths and air temperature parameters were evaluated by simple correlation analysis. Statistical analysis was performed using JMP Pro version 14 (SAS Institute Inc., Cary, NC, USA).

## 3. Results

The air temperature profiles in Gifu prefecture, Japan, are summarized in Table 1. The mean air temperature was 16.3 ± 8.3 ℃. The highest and lowest air temperatures were 26.9 ± 8.1 ℃ and 7.1 ± 8.5 ℃, respectively. The number of deaths stratified by cause in Gifu prefecture, Japan, is shown in Table 2. The total number of deaths was 2.62 ± 0.48 (/one hundred thousand people/day). The causes of death in Gifu prefecture were (1) malignant neoplasm, (2) heart disease, (3) cerebrovascular disease, (4) pneumonia, (5) senility, (6) accidents, (8) suicide, (9) renal failure, and (10) vascular and unspecified dementia, in this order.

Next, we compared the number of deaths stratified by cause, as shown in Table 3. The mean air temperature in January (winter season) was the lowest among the months. The number of deaths in January (winter season) was the highest among the months, and the main causes of death were heart disease, cerebrovascular disease, pneumonia, accidents, and renal failure. The total number of deaths in January (winter season) was significantly higher than that from May to October. The number of deaths due to heart disease in January (winter season) was significantly higher than that from March to December. The number of deaths due to cerebrovascular disease and pneumonia in January (winter season) was significantly higher than that from April to October. The number of deaths due to accidents in January (winter season) was significantly higher than that from March to November. The number of deaths due to renal failure in January (winter season) was significantly higher than that from March to October.

The simple correlation analysis of the number of deaths stratified by cause and mean air temperature is shown in Table 4. Negative relationships were noted between the mean air temperature and heart disease, cerebrovascular disease, senility, pneumonia, accidents, and renal failure. In addition, we further analyzed the number of deaths stratified by cause and all air temperature parameters in both sexes (Table 5). The number of deaths due to heart disease, cerebrovascular disease, senility, pneumonia, accidents, or renal failure was significantly and negatively related to all air temperature parameters in both men and women. Regarding death by suicide, the mean of the highest air temperature and the highest air temperature were significantly and positively correlated with the number of deaths only in women. Regarding vascular and unspecified dementia, the mean air temperature, mean of the highest air temperature, and the lowest air temperature were negatively associated with the number of deaths only in men.

## 4. Discussion

In this study, we evaluated the relationship between the number of deaths stratified by cause and air temperature parameters in Gifu prefecture, Japan. The number of deaths due to heart disease, cerebrovascular disease, senility, pneumonia, accidents, and renal failure was significantly negatively correlated with air temperature parameters.

There have been many reports on the relationship between the number of deaths and air temperature. Analitis et al. reported that a 1 ℃ decrease in air temperature corresponds to a 1.35% increase in the total number of deaths, and a 1.72%, 3.3%, and 1.25% increase in the number of deaths due to cardiovascular disease, respiratory disease, and cerebrovascular disease in 15 European cities by using data from 1990–2000 [5]. Wanitschek et al. also found that a 7.5 ℃ increase in air temperature in winter reduces the incidence of acute cerebrovascular disease [4]. In Japan, the incidence of acute myocardial infarction is closely associated with lower air temperature, especially lower than 10 ℃ [9]. Chaochen et al. reported that the temperature variability was closely associated with senility [26]. In addition, we previously confirmed that the incidence of accidental death, including asphyxia [27], drowning [28], falls [29], cold [30], and fire [31], is closely associated with lower air temperature. In this study, we found a significant relationship between air temperature parameters and the number of deaths due to heart disease, cerebrovascular disease, senility, pneumonia, and accidents in Gifu prefecture, as suggested in previous reports. Although Nakaji et al. previously reported a relationship between the number of deaths by cause and air temperature and noted seasonality in the number of deaths due to infection, diabetes mellitus, heart disease, cerebrovascular disease, digestive disorders, and suicide using data from 1970 to 1999 [21], their results may not be the same as ours because of the marked aging of society in Japan. 

In this study, we found a significant relationship between the number of deaths due to renal failure and air temperature parameters. Phillips et al. reported that the incidence and mortality of acute renal failure were significantly higher between January and March [32]. The mortality due to end-stage renal failure was the highest in January [33]. In Japan, Iwagami et al. reported that the incidence of acute renal failure was higher in winter, resulting in a serious state [34]. Iseki et al. also found that end-stage renal failure was closely associated with intradiurnal temperature change [35]. In this study, there was a significant relationship between the number of deaths due to renal failure and air temperature in Gifu prefecture, Japan. It is well known that blood pressure in winter is higher than that in summer [36]. In addition, the incidence of heart disease, cerebrovascular disease, and infectious disease, including pneumonia, is higher in winter than in summer [4,5,6,7,8,10,11,12,13,14,15,37]. These factors may affect the relationship between the number of deaths due to renal failure and air temperature in Gifu prefecture, Japan. Therefore, patients with renal failure require closer attention in winter to prevent deterioration and to improve their health status. 

There are several limitations. First, this was an ecological study and individual data were not available and many factors, including age and physical factors, were not evaluated. The air temperature would affect many aspects including physical activity, sleeping, and consumption of energy and nutrients resulting in an increase of the number of deaths. Second, we evaluated the relationship between the number of deaths stratified by cause and air temperature parameters only in Gifu prefecture, Japan. Thus, the results of this study may not be applicable to other prefectures in Japan. 

In conclusion, we found a significant relationship between the number of deaths, including that due to renal failure and air temperature parameters in Gifu prefecture, Japan. 

## Figures and Tables

**Table 1 healthcare-08-00035-t001:** Climate parameters from 2007 to 2016 in Gifu prefecture, Japan.

Number of months	120		
Mean air temperature (℃)	16.3	±	8.3
Mean of the highest air temperature (℃)	21.1	±	8.4
Mean of the lowest air temperature (℃)	12.1	±	8.5
The highest air temperature (℃)	26.9	±	8.1
The lowest air temperature (℃)	7.1	±	8.5

Mean ± standard deviation (SD).

**Table 2 healthcare-08-00035-t002:** Number of deaths stratified by cause in Gifu prefecture, Japan.

Cause	All			Men			Women		
Total	2.62	±	0.48	2.96	±	0.34	2.82	±	0.95
Malignant neoplasm	0.77	±	0.05	0.95	±	0.07	0.61	±	0.06
Heart disease	0.45	±	0.10	0.44	±	0.10	0.47	±	0.10
Cerebrovascular diseases	0.27	±	0.04	0.26	±	0.04	0.27	±	0.04
Pneumonia	0.25	±	0.04	0.29	±	0.05	0.22	±	0.04
Senility	0.17	±	0.06	0.10	±	0.04	0.24	±	0.08
Accidents	0.11	±	0.02	0.12	±	0.03	0.09	±	0.03
Suicide	0.06	±	0.01	0.08	±	0.02	0.03	±	0.01
Renal failure	0.05	±	0.01	0.06	±	0.02	0.05	±	0.02
Vascular and unspecified dementia	0.02	±	0.01	0.01	±	0.01	0.03	±	0.01

Mean ± SD; /one hundred thousand people/day.

**Table 3 healthcare-08-00035-t003:** Comparison of the number of deaths per month in Gifu prefecture, Japan.

Month	Total				Malignant Neoplasm				Heart Diseases				Cerebrovascular Diseases				Senility		
January	3.06	±	0.43		0.77	±	0.05		0.63	±	0.06		0.31	±	0.03		0.20	±	0.06
February	2.90	±	0.53		0.78	±	0.04		0.56	±	0.06		0.31	±	0.03		0.20	±	0.07
March	2.76	±	0.49		0.75	±	0.06		0.51	±	0.05	a	0.29	±	0.03		0.17	±	0.06
April	2.63	±	0.43		0.77	±	0.05		0.47	±	0.04	a	0.27	±	0.02	a	0.17	±	0.05
May	2.50	±	0.39	a	0.75	±	0.05		0.41	±	0.02	a	0.26	±	0.03	a	0.15	±	0.05
June	2.31	±	0.34	a	0.75	±	0.04		0.36	±	0.02	a	0.23	±	0.02	a	0.14	±	0.05
July	2.32	±	0.37	a	0.76	±	0.05		0.36	±	0.03	a	0.23	±	0.02	a	0.15	±	0.05
August	2.37	±	0.40	a	0.78	±	0.04		0.37	±	0.03	a	0.24	±	0.02	a	0.14	±	0.04
September	2.38	±	0.37	a	0.79	±	0.06		0.36	±	0.03	a	0.22	±	0.02	a	0.17	±	0.06
October	2.53	±	0.41	a	0.78	±	0.05		0.40	±	0.03	a	0.25	±	0.01	a	0.19	±	0.07
November	2.78	±	0.50		0.79	±	0.05		0.48	±	0.05	a	0.29	±	0.03		0.20	±	0.05
December	2.90	±	0.50		0.78	±	0.08		0.55	±	0.05	a	0.29	±	0.02		0.20	±	0.06
	Pneumonia				Accidents				Renal failure				Suicide				Vascular and unspecified dementia		
January	0.32	±	0.03		0.14	±	0.02		0.07	±	0.01		0.05	±	0.01		0.02	±	0.01
February	0.30	±	0.02		0.12	±	0.03		0.06	±	0.01		0.06	±	0.01		0.02	±	0.01
March	0.28	±	0.01		0.11	±	0.02	a	0.06	±	0.01	a	0.06	±	0.01		0.02	±	0.01
April	0.26	±	0.01	a	0.10	±	0.01	a	0.06	±	0.01	a	0.06	±	0.01		0.02	±	0.01
May	0.25	±	0.03	a	0.09	±	0.02	a	0.05	±	0.01	a	0.06	±	0.01		0.01	±	0.01
June	0.21	±	0.02	a	0.08	±	0.01	a	0.05	±	0.01	a	0.05	±	0.01		0.01	±	0.01
July	0.20	±	0.03	a	0.09	±	0.02	a	0.05	±	0.01	a	0.06	±	0.01		0.02	±	0.01
August	0.21	±	0.02	a	0.10	±	0.01	a	0.05	±	0.01	a	0.06	±	0.01		0.02	±	0.01
September	0.20	±	0.02	a	0.08	±	0.01	a	0.04	±	0.01	a	0.06	±	0.01		0.02	±	0.01
October	0.24	±	0.02	a	0.10	±	0.01	a	0.05	±	0.01	a	0.06	±	0.01		0.02	±	0.01
November	0.28	±	0.02		0.10	±	0.02	a	0.06	±	0.01		0.06	±	0.01		0.02	±	0.01
December	0.28	±	0.03		0.13	±	0.01		0.06	±	0.01		0.04	±	0.01		0.02	±	0.01

Mean ± SD; a: *p* < 0.05 vs January.

**Table 4 healthcare-08-00035-t004:** Simple correlation analysis of number of deaths (/one hundred thousand people/day) and mean air temperature in Gifu prefecture, Japan.

	r	*p*	Single Regression (Y: Number of Deaths, X: Climate Parameters)
Total	−0.509	<0.001	Y = − 0.0295X + 3.0995
Malignant neoplasm	−0.037	0.690	Y = − 0.0002X + 0.7750
Heart disease	−0.885	<0.001	Y = − 0.0104X + 0.6240
Cerebrovascular diseases	−0.790	<0.001	Y = − 0.0034X + 0.3211
Senility	−0.303	<0.001	Y = − 0.0022X + 0.2077
Pneumonia	−0.855	<0.001	Y = − 0.0046X + 0.3267
Accidents	−0.687	<0.001	Y = − 0.0020X + 0.1384
Renal failure	−0.627	<0.001	Y = − 0.0015X + 0.0698
Suicide	0.067	0.466	Y = 0.0001X + 0.0543
Vascular and unspecified dementia	−0.147	0.109	Y = − 0.0001X + 0.0212

**Table 5 healthcare-08-00035-t005:** Relationship between number of deaths stratified by cause and air temperature parameters according to simple correlation analysis.

Air Temperature Parameters	Total	Malignant Neoplasm	Heart Disease	Cerebrovascular Diseases	Senility	Pneumonia	Accidents	Renal Failure	Suicide	Vascular and Unspecified Dementia
All										
Mean air temperature (℃)	−0.51	−0.03	−0.92	−0.78	−0.31	−0.86	−0.69	−0.63	0.10	−0.15
Mean of the highest air temperature (℃)	−0.51	−0.04	−0.92	−0.78	−0.32	−0.85	−0.69	−0.63	0.12	−0.16
Mean of the lowest air temperature (℃)	−0.51	−0.02	−0.91	−0.79	−0.29	−0.86	−0.67	−0.62	0.08	−0.14
The highest air temperature (℃)	−0.52	−0.05	−0.88	−0.77	−0.30	−0.84	−0.72	−0.63	0.13	−0.13
The lowest air temperature (℃)	−0.50	−0.05	−0.50	−0.76	−0.32	−0.85	−0.63	−0.61	0.09	−0.17
Men										
Mean air temperature (℃)	−0.81	−0.06	−0.89	−0.70	−0.32	−0.79	−0.60	−0.51	−0.22	−0.18
Mean of the highest air temperature (℃)	−0.82	−0.08	−0.89	−0.70	−0.33	−0.79	−0.60	−0.51	−0.23	−0.19
Mean of the lowest air temperature (℃)	−0.81	−0.05	−0.89	−0.71	−0.31	−0.80	−0.58	−0.50	−0.22	−0.17
The highest air temperature (℃)	−0.82	−0.10	−0.89	−0.68	−0.32	−0.78	−0.62	−0.50	−0.22	−0.17
The lowest air temperature (℃)	−0.79	−0.07	−0.85	−0.69	−0.33	−0.79	−0.54	−0.50	−0.23	−0.20
Women										
Mean air temperature (℃)	−0.22	0.02	−0.88	−0.69	−0.29	−0.74	−0.62	−0.53	0.17	−0.11
Mean of the highest air temperature (℃)	−0.22	0.02	−0.88	−0.68	−0.30	−0.74	−0.62	−0.54	0.18	−0.12
Mean of the lowest air temperature (℃)	−0.21	0.02	−0.88	−0.68	−0.27	−0.74	−0.60	−0.52	0.15	−0.09
The highest air temperature (℃)	−0.21	0.02	−0.89	−0.68	−0.28	−0.73	−0.64	−0.54	0.20	−0.09
The lowest air temperature (℃)	−0.21	0.00	−0.85	−0.66	−0.30	−0.72	−0.58	−0.51	0.16	−0.13

Bold values: *p* < 0.05 by simple correlation analysis.

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
