# Peer review of "Relationship between Air Temperature Parameters and the Number of Deaths Stratified by Cause in Gifu Prefecture, Japan"

_healthcare, 2020, doi:10.3390/healthcare8010035_

Round 1
Reviewer 1 Report
In the Abstract, it is important to point out whether higher or lower temperature was associated with different causes of death.The introduction is very short. The authors stated that they had reported a significant relationship between air temperature and the number of deaths in many cities of Japan. Did these studies also investigate the number of deaths stratified by causes? The knowledge gap for research should be clarified further.
For Table 3, it would be helpful to list the mean temperature by month so that international readers can understand the seasons (e.g. when is summer or winter?) in Japan.
In Table 4 and 5, it is essential to indicate in the main text whether higher or lower temperature was associated with a specific cause of death. Was there direction of relationship (positive or negative) indicated by the correlation analysis?
The discussion has indicated the whether the association was with a higher or lower temperature. This was quite well written. The same approach of description should be used in the results. This would make more sense for the interpretation.
Author Response
Dear Author
Thank you for the thoughtful and constructive feedback you provided regarding our manuscript, Relationship between Air Temperature Parameters and the Number of Deaths Stratified by Cause in Gifu Prefecture, Japan (healthcare-710132) .
We hereby resubmit our manuscript for a secondary evaluation.
Thank you once again for your consideration of our paper.
Sincerely,
Kind Regards,

Reviewer 2 Report
Dear Authors,
This is an interesting study, but while reading the manuscript I had some comments and questions:
Abstract
Information on the average temperature and the number of deaths from the entire study period, i.e. 10 years, does not refer to the purpose of the work. What's more, further information about the highest number of deaths in January also adds nothing, because the reader does not know whether the month also significantly differs from others in terms of temperature (is it the coldest? The most humid?). The question also arises: if the relationship between the number of deaths and temperature was found for several diseases or causes (accidents), why in the conclusion was only renal failure named?
Introduction
I do not fully understand the intentions of the first sentence regarding the increasing number of deaths as a consequence of population aging in the context of the title / purpose of the study? This would make sense if the effect of air temperature on the number of deaths depended on age? But the text lacks information on this?
Definitely more information is needed to introduce the reader to the topic. What do the authors mean by "lower temperature" (line 42)?In different regions of the earth this may have a completely different meaning, and readers may not necessarily know the details of the climate in Japan.
I am also thinking about possible strategies to prevent death affected by temperature. Could the authors develop this thought more?
Methods
Why were such causes of death chosen? Were the temperature relationships somehow proven for these deaths? This wonders the reader, especially in the case of deaths due to accidents (traffic accidents? others?)?
Results
The way the results are presented does not emphasize the relationship between temperature and the number of deaths, maybe it shows more the relationship between the month and the number of deaths. The reader does not know what the typical / average temperatures are in each month.
Discussion
Line 123: The example of the effect of a 1 degree reduction in temperature on the number of deaths should have a reference, i.e. at what temperature it was observed. The reader may have doubts here.
The authors made very little reference to the possible causes of the observed relationships as well as to the limitations of the study.
The season (including air temperature) affects many aspects that may affect human health. It can condition e.g. physical activity, sleep duration, time spent in the air, consumption of energy and nutrients.
Author Response

(The authors gave the same response as above.)

Reviewer 3 Report
Thank you for the opportunity to review this paper. Overall, I found it to be interesting and very well presented. Some minor comments below.
The introduction would benefit from further supporting literature and a more comprehensive background. I would like to know why it is important to know that air temperature effects the rate of death and some further information about the meteorological fluctuations in Japan (and globally perhaps?). Line 37: I would slightly rephrase to say "The number of deaths in Japan have been markedly increasing....". And perhaps give a little more detail here rather than jumping into health outcomes. Some further literature about the effects of air temp on specific diseases may also provide more context. When you are talking about number of deaths - does this include all ages - it might be worth defining this in your methods. Are the air temps in Gifu prefecture transferable to other areas in Japan? It was great to see you have identified the limitations clearly. It might be worth looking at some case studies in the future where you can assess a variety of variables.
Author Response

(The authors gave the same response as above.)

Reviewer 4 Report
This is an excellent contribution on plausible causal connection between air temperature and related NCDs mortality such COPD in a defined Japanese prefecture.
It is right in the center of environmental focus of this journal in public health.
Methodologically well funded this paper is worthy of publishing.
Yet I believe minor strengthening of evidence base with some convenient sources outside Japanese academic literature might help support claims in the text and make it more reliable.
Therefore I suggest few sources beneath but authors are free to add few others at their own will:
Ogura, S., & Jakovljevic, M. (2014). Health Financing Constrained by Population Aging-An Opportunity to Learn from Japanese Experience. Serbian Journal of Experimental and Clinical Research, 15(4), 175-181.
Jakovljevic, M. B., Nakazono, S., & Ogura, S. (2014). Contemporary generic market in Japan–key conditions to successful evolution. Expert review of pharmacoeconomics & outcomes research, 14(2), 181-194.
Soriano, J. B., Abajobir, A. A., Abate, K. H., Abera, S. F., Agrawal, A., Ahmed, M. B., ... & Alam, N. (2017). Global, regional, and national deaths, prevalence, disability-adjusted life years, and years lived with disability for chronic obstructive pulmonary disease and asthma, 1990–2015: a systematic analysis for the Global Burden of Disease Study 2015. The Lancet Respiratory Medicine, 5(9), 691-706.
Jakovljevic, M., Jakab, M., Gerdtham, U., McDaid, D., Ogura, S., Varavikova, E., ... & Getzen, T. E. (2019). Comparative financing analysis and political economy of noncommunicable diseases. Journal of medical economics, 22(8), 722-727. World Health Organization. (2019). Sixth Regional Workshop on Leadership and Advocacy for the Prevention and Control of Noncommunicable Diseases (LeAd-NCD), Saitama, Japan, 12-15 March 2019: meeting report. Manila: WHO Regional Office for the Western Pacific.
Author Response

(The authors gave the same response as above.)

Round 2
Reviewer 1 Report
The authors have responded well to the questions of the reviewer and made significant improvement to the manuscript.
Reviewer 2 Report
Dear Authors,
I appreciate the changes made to the manuscript. I think it strengthened the paper.
However, I still need to get closer to the topic in the Introduction. I would like more information on the impact of temperature or other parameters on the risk of death (e.g. as other authors explain their observations).